# The Association between Plasma Concentration of Phytoestrogens and Hypertension within the Korean Multicenter Cancer Cohort

**DOI:** 10.3390/nu13124366

**Published:** 2021-12-05

**Authors:** Juyeon Lee, Ju-Young Kang, Kwang-Pil Ko, Sue-Kyung Park

**Affiliations:** 1Department of Preventive Medicine, College of Medicine, Seoul National University, 103 Daehakro, Jongnogu, Seoul 03080, Korea; juyeon87@snu.ac.kr; 2Department of Biomedical Science, College of Medicine, Seoul National University, 103 Daehakro, Jongnogu, Seoul 03080, Korea; 3Department Cancer Institution, Seoul National University, 103 Daehakro, Jongnogu, Seoul 03080, Korea; 4Department of Medicine, College of Medicine, Seoul National University, 103 Daehakro, Jongnogu, Seoul 03080, Korea; zoungki82@naver.com; 5Department of Preventive Medicine, Gachon University College of Medicine, 38-13 Dokjeom-ro 3beon-gil, Namdong-gu, Incheon 21565, Korea; kpko@gachon.ac.kr; 6Integrated Major in Innovative Medical Science, College of Medicine, Seoul National University, 103 Daehakro, Jongnogu, Seoul 03080, Korea

**Keywords:** hypertension, prehypertension, isoflavone, phytoestrogen, soybean, equol, enterolactone

## Abstract

In order to examine the association between plasma phytoestrogen concentration (genistein, daidzein, equol and enterolactone) and hypertension, we conducted a nested case–control study for 229 hypertension cases including 112 prehypertension and 159 healthy controls derived from the Korean Multi-center Cancer Cohort (KMCC). The concentration of plasma phytoestrogens was measured using time-resolved fluoroimmunoassay. We assessed the association between plasma phytoestrogens and hypertension using logistic regression models using odds ratio (OR) and 95% confidence interval (95%CI). The highest tertile of plasma equol and enterolactone concentration exhibited a significantly decreased risk of hypertension (equol, OR = 0.34, 95%CI 0.20–0.57; enterolactone, OR = 0.32, 95%CI 0.18–0.57), compared with the lowest tertile. Equol and enterolactone showed reduced ORs for prehypertension (the highest tertile relative to the lowest tertile, OR = 0.50, 95%CI 0.26–0.96; OR = 0.38, 95%CI 0.19–0.75, respectively) and hypertension (OR = 0.42, 95%CI 0.22–0.81; OR = 0.28, 95%CI 0.14–0.54, respectively). There was a stronger association in hypertension (the highest tertile relative to the lowest tertile in obesity vs. non-obesity; equol, OR = 0.06 vs. 0.63; enterolactone, OR = 0.07 vs. 0.46; both *p*-heterogeneity < 0.01). This study suggests that equol and enterolactone may contribute to prevent primarily prehypertension and hypertension, and control cardiovascular disease (CVD) based on the continuum of hypertension and CVD. Further study to assess hypertension risk based on useful biomarkers, including phytoestrogens, may contribute to primary prevention of hypertension.

## 1. Introduction

Hypertension has been referred as an epidemic due to its high prevalence worldwide and it has been predicted that 29.5% of the global adult population worldwide will be hypertension patients by 2025 [1]. In Korea, the number of people with hypertension has increased steadily, and is now over 12.0 million [2,3]. The increase in Koreans eating a Westernized diet containing high-sodium and fattening westernized foods, in addition to traditional Korean salty foods, has been noted as one of the main risk factors contributing to this rise in hypertension [4]. However, epidemiological studies on the role of dietary habits on hypertension risk in Korea are limited.

Soy is structurally similar to 17 β-estradiol, the primary female sex hormone. It is known that soy will improve menopausal-related health outcomes related to bone resorption and urinary incontinence [5,6,7]. In addition, soy may play a role similar to phytoestrogens in binding to estrogen receptors and therefore interfering with the action of estrogen, which is a well-established risk factor for hormone-dependent diseases such as prostate and breast cancers [8,9]. Additionally, anti-inflammatory and antioxidant effect of soy intake may have a protective effect for non-hormone dependent diseases [10,11].

Phytoestrogens are estrogen-like compounds derived from plants. Four phenolic compounds classified as phytoestrogens are isoflavones, lignans, stilbene, and coumestan [12,13]. Isoflavones and lignans are the major classes of phytoestrogen and their health benefits have been widely studied [14,15,16,17].

The major isoflavones are genistein and daidzein. The glycoside forms of these components are digested in the human intestinal wall and metabolized in a biologically active form [18]. Genistein has been studied for its antidiabetic effect, hypolipemic effect, cancer chemo-preventive effect, anti-inflammatory effect and modulation of adipose tissue function [19,20,21]. Daidzein is the second abundant isoflavone in soy. It has been shown to have an antidiabetic effect by stimulating insulin secretion [22] and nitric oxide production and improving endothelial function (EF), suggesting that soy plays a role in preventing hypertension [23]. Equol is produced from daidzein by gut microbes [24], and is known to influence the clinical effectiveness of soy intake according to its producing status [25]. The lignan is bioactive and non-caloric phenolic plant compounds. Its weak estrogenic properties and other biochemical properties suggest that it has potential for nutritional significance in the prevention of chronic disease [26].

Previous studies have reported an association between soy isoflavone intake and hypertension in Korea [27,28]. However, isoflavone intake level was estimated by using inaccurate methods such as a food frequency questionnaire and dietary recall, which are prone to underestimation of isoflavone intake. Thus, measuring phytoestrogen levels and employing a prospective cohort study design may help in reducing such recollection and misclassification bias [29].

In contrast to prior studies which used relatively inaccurate methods of evaluating dietary exposure, such as estimation based on a questionnaire [30], our study focused on accurate quantitative measurement of the dietary exposure in serum level and detailed analysis for each of the specific types of compounds which make up phytoestrogen.

Therefore, in this study, we conducted the associations of the concentration of phytoestrogen levels in plasma, a marker of dietary isoflavone, with hypertension in Korea. We also investigated the associations of serum phytoestrogen levels with hypertension according to obesity.

## 2. Materials and Methods

### 2.1. Study Population

This study was based on a nested case–control study within the Korean Multicenter Cancer Cohort (KMCC). In total, 19,688 subjects were recruited from 1993 to 2004 from 4 rural and urban areas (Haman, Chungju, Uljin, and Youngil) in Korea. Information on environmental factors and lifestyle factors were obtained from a questionnaire conducted by well-trained interviewers. Anthropometric indices such as height, weight, waist circumference and hip circumference were also measured. Detailed information containing study design and protocols are described elsewhere [31].

Subjects were selected from the KMCC in this population-based nested case–control study. Among 1122 subjects, we excluded those with prior cancer history before enrollment or new cancer development (*n* = 433), those with a history of anti-hypertensive drugs (*n* = 137), and those with insufficient plasma (<200 μL) (*n* = 164). Finally, a total 229 hypertension (Incident cases with hypertension and pre-hypertension) and 159 healthy control groups were selected (Figure 1). All participants provided written informed consent to participate, and the study protocols were approved by the institutional review board (IRB) of Seoul National University Hospital (IRB No: 0110-084-002).

### 2.2. Assessment of Hypertension

Anthropometric measurements have been conducted following standard procedure. Arterial blood pressure was measured in sitting position with at least 5 min at rest at the end of the physical examination. Because of the possibility of differences in blood pressure measurement, the measurements were taken twice on the right arm with it relaxed and well supported by a table, with an angle of 45° from the trunk. A mean of the last two measurements was considered for inclusion in the database [31].

All study subjects were selected as those who had no history of cancers at the time of cohort enrollment and had never taken prior antihypertensive drugs. Study participants were classified into three groups, such as hypertension cases, prehypertension cases, and healthy controls. Subjects with SBP in 120–139 mm/Hg or DBP in 80–89 mm/Hg were classified to prehypertension, while those who had SBP ≥ 140 mm/Hg or DBP ≥ 90 mm/Hg were classified as hypertension. The high blood pressure (BP) group was defined to include both prehypertensive patients and hypertensive patients. Healthy controls were selected as those with SBP < 120 mm/Hg and DBP < 80 mm.

### 2.3. Measurement of Plasma Concentration of Phytoestrogens

Blood samples were collected at the base-line and stored in −70 °C with buffer, glucuronidase and sulfatase treated. Blood samples were incubated at 37 °C overnight for preparation. A time-resolved fluoroimmunoassay kit (Labmaster, Finland) was used to measure the plasma phytoestrogens (isoflavones (genistein, daidzein and equol) and lignans (enterolactone)) samples, which is a reliable method for measuring plasma phytoestrogen level. The correlation coefficient between measurement of the time-resolved fluoroimmunoassay and gas chromatography-mass-spectrometry was high; genistein 0.96, daidzein 0.95 and equol 0.98 [32,33]. Free genistein was extracted twice with diethyl ether and dried in a water bath. The residue was diluted in 200 μL of assay buffer and 20 μL was incubated prewashed goat anti-rabbit IgG-coated microtitration wells. Then, 100 μL of antigenistein antibody working solution and genistein–Eu tracer working solution were added to the wells. Plates were shaken gently for 90 min at room temperature and washed four times. After 200 μL of an enhancement solution was treated and the plates were shaken for 5 min, time-resolved fluoroimmunoassay was measured using the Victor 3 1420 Multilabel Counter (Perkin-Elmer, Turku, Finland). Daidzein, equol and enterolactone were measured in a similar way so that the pretest for phytoestrogen measurement was only implemented in genistein. Additionally, coefficients of variation (CV) mean and SD were calculated in each of phytoestrogen levels (4.70 ± 2.87% for genistein, 3.66 ± 2.85% for daidzein, 3.68 ± 2.60% for equol, and 4.22 ± 2.64% for enterolactone) and the result was within the recommended CV% range (typically <10% over the standard curve range) of manufacturer’s manual.

### 2.4. Statistical Analyses

To compare the general characteristics of subjects with total cases, including hypertension and prehypertension, and healthy controls, we conducted the chi-square test for categorical variables and an independent t-test for continuous variables. We divided the plasma phytoestrogens (genistein, daidzein, equol, and enterolactone) levels into tertiles and used them in further analysis.

In this study, the association with plasma phytoestrogens levels on the risk of hypertension and prehypertension were evaluated by calculating odds ratios (OR) and 95% confidence intervals (CI) using the logistic regression model, and adjusted for seven potential confounders including age, sex, enrollment year, education, smoking history, alcohol drinking history, and obesity. P-trend was calculated to assess the dose-dependent association.

Polytomous logistic regression analysis with adjustment for age, sex, enrollment year, education, smoking history, and alcohol drinking history was performed to evaluate the association between plasma phytoestrogens levels and the risk of hypertension and prehypertension according to obesity status (body mass index < 25 kg/m^2^ vs. ≥25 kg/m^2^).

We also assessed the collinearity between independent variables using the variance inflation factor (VIF) and the Pearson’s correlation coefficient. All statistical analyses were performed using SAS version 9.4 (SAS Institute Inc., Cary, NC, USA).

## 3. Results

### 3.1. General Characteristics

Baseline characteristics of the study population are described in Table 1. There was no difference in mean age (standard deviation); 62.5 (7.5), 63.9 (8.9) in healthy controls and total cases including hypertension and prehypertension, respectively. Compared with the healthy control group, the high BP group had higher BMI and a history of drinking alcohol. None of the characteristics such as sex, enrollment year, educated years, diabetes history, and smoking history showed significant difference among the groups.

### 3.2. Association between Plasma Phytoestrogen Levels and High BP including Both Prehypertension and Hypertension

Table 2 shows the association of plasma phytoestrogen levels with high BP including both prehypertension and hypertension. The highest tertiles of plasma concentrations of equol and enterolactone, respectively, were associated with a reduced risk for high BP (OR = 0.34, 95% CI = 0.20–0.57 for equol; OR = 0.32, 95% CI = 0.18–0.57 for enterolactone). The increase in plasma concentration of equol and enterolactone was associated with a continual decrease in the OR (*p*-trend < 0.01, respectively). Plasma concentration of genistein and daidzein levels had no significant association with high BP.

### 3.3. Association with Plasma Phytoestrogen Levels for Hypertension and Prehypertension, Respectively

Table 3 shows the association of plasma phytoestrogen levels with each hypertension and prehypertension group, respectively. Highest tertile of plasma concentration of equol and enterolactone had inverse association with both prehypertension (OR = 0.50, 95% CI = 0.26–0.96 for equol; OR = 0.38, 95% CI = 0.19–0.75 for enterolactone) and hypertension (OR = 0.42, 95% CI = 0.22–0.81 for equol; OR = 0.28, 95% CI = 0.14–0.54 for enterolactone). Both equol and enterolactone showed a stronger association with hypertension than that with prehypertension (*p*-ordinal = equol < 0.01; enterolactone < 0.01). Plasma concentration of genistein and daidzein levels had no significant association with prehypertension and hypertension.

The multivariate model showed goodness of fit (Hosmer-Lemeshow statistic), as well as partial coefficient regression and significance between phytoestrogens and pre-hypertension/hypertension (Appendix A).

### 3.4. Association between Plasma Phytoestrogen Levels and Hypertension According to Obesity

A stratified analysis by obesity (BMI < 25 kg/m^2^, ≥25 kg/m^2^) result is presented in Table 4. The association with plasma equol and enterolactone levels in the obese group was much stronger than that in non-obese group (the highest tertiles of equol and enterolactone in the obese group, OR = 0.06, 95% CI 0.01–0.35; OR = 0.07, 95% CI 0.13–0.34; the highest tertiles in non-obese group, OR = 0.63, 95% CI 0.34–1.12; OR = 0.46, 95% CI 0.24–0.88; *p*-heterogeinity < 0.01 and *p*-heterogeinity < 0.01, respectively).

## 4. Discussion

Plasma concentration of equol and enterolactone had a dose-dependent inverse association with risk of hypertension and prehypertension, especially in the obese group. Both equol and enterolactone had a stronger association with hypertension than with prehypertension. Plasma concentration of genistein and daidzein levels had no significant association with hypertension and prehypertension.

Existing studies on the association between isoflavones and hypertension have consistently reported that soy isoflavone intake reduces the risk of hypertension or reduces BP [34,35], which was also confirmed in several clinical trials and a meta-analysis of clinical trials [36]. However, the effects between SBP and DBP were not consistent in those studies. The SBP reduction effect of isoflavones was consistent, regardless of study design [34,36,37], whereas the DBP reduction effect was not clear (DBP reduction [37]; no effect [36]). Furthermore, prior studies did not investigate whether specific isoflavone metabolites act differently on vascular function because they measured isoflavones with a dietary questionnaire. One of the reasons for using isoflavone dietary intake as a major exposure factor in previous studies may be the vulnerability to bias of dietary questionnaire measurement methods and the estimation error or underestimation of specific isoflavone metabolites [38,39].

The main sources of isoflavones are derived mainly from beans and their products and legume seeds [40]. Ingestion of dietary isoflavones is metabolized from genistein to daidzein in the liver and then metabolized to equol in the colon [41]. Equol metabolism is related to whether the colon has a specific microbiopta that metabolizes equol. Accordingly, some people produce equol and some do not, and the level of equol varies from person to person [42]. This background suggests that inter-individual differences in equol production play a substantial role in hypertension control and blood pressure control, which supports our results.

Two clinical trials published in 2015 provided indirect results supporting the “Equol producer hypothesis” as a modulator of BP. In the first trial, a crossover experiment in postmenopausal women was perfomed to compare soy diet intervention to a non-soy control diet. As a result, it was confirmed that there was an effect on the reduction in DBP, which was significantly reduced only in equol producers (7.7% reduction for women with metabolic syndome; 3.3% reduction for women without metabolic syndrome) [43]. In the other clinical trial in postmenopausal women with prehypertension or untreated hypertension, two interventions of whole soy diet and daidzein (a precursor of equol) diet were compared to the control diet. As a result, there was no 6-month difference in ambulatory blood pressure and brachial blood flow-mediated diastolic changes between the three groups, even in the equol producer. [44]. In order to obtain the direct evidence of equol intervetion on hypertension, a clinical trial was started from 2015 in China, the results were not reported yet [45].

To the best of our knowledge, the evidence for the antihypertensive effect of phytoestrogen by altering renin-angiotensin-aldosterone system (RAAS) was still controversial to some extent. In one rat model study, genistein suppressed the expression of the angiotensin II type 1 receptor in the endothelial cells [46]. However, other studies failed to demonstrate the significant change in RAAS by genistein in animal models [47,48]. As for daidzein, it seems to counteract the increase in blood pressure induced by angiotensin I by inhibiting ACE activity in endothelial cells [49,50]. Evidence on direct association between phytoestrogens and catecholamine was limited. Catecholamine raise intracellular calcium level in vascular smooth muscle cells by opening receptor-operated calcium channels (ROCC), activating calmodulin and myosin light chain kinase, and opening of L-type-voltage-gated calcium channel (VGCC). However, phytoestrogens such as biochanin A and daidzein seem to counter act the catecholamine by inhibiting VGCC or ROCC [51,52].

Our finding for equol is biologically plausible. Equol is an estrogen-like substance that can induce ER target gene expression more strongly than genistein and daidzein, and it is the most powerful isoflavone among isoflavones [53,54]. The effect of the reduction in BP or hypertension risk by equol may be due to its antioxidant and calcium antagonistic potencies. Equol is participated in nitric oxide (NO) generation by the rapid activation of eNOS and its binding to HSP90, resulting indirectly in vasorelaxation [55]. Another mechanism is that equol is involved in vasodilation by acting directly on vascular smooth muscle through the calcium channels as a calcium antagonist [56].

A similar result to ours for a stronger association in the obese group was also found in a meta-analysis of randomized controlled trials. The study confirmed that the effect for BP reduction was better in the group with high BMI (≥28 kg/m^2^) than the other BMI group [57]. It is biologically plausible, implying the biologic mechanism of interventional function of phytoestrogen in the progress of obesity to hypertension. Obesity is known to induce hypertension through the mechanisms in stimulating the renin-angiotensin system and in increasing inflammatory response and oxidative stress via NADPH oxidase activation [58,59,60]. The processes leads to cause hypertension through endothelial dysfunction and vasoconstriction by nitrogen oxide reduction [61,62,63].

Several studies noted that prostate cancer progression or its aggressiveness has been associated with obesity and hypertension [64,65]. On the other hand, phytochemicals (including phytoestrogen) have shown to exert an anti-cancer or cancer-prevention effect against prostate cancer [66]. The underlying mechanism of how phytochemicals reduce the risk of prostate cancer progression may be explained not only by the fact that phytoestrogen interfere with endogenous estrogen which is a risk factor for prostate cancer, but also by the fact that phytoestrogen is associated with decreased likelihood of developing hypertension and obesity. There seems to be a complex network of influences among the risk factors for prostate cancer such as estrogen, hypertension, obesity and protective dietary factors for prostate cancer including vitamins, phytochemicals and phenolic compounds, which are known to have an anti-cancer effect [66].

Another important finding of ours is that the lignan enterolactone is associated with a reduction in hypertension. However, two prior studies did not investigate which specific lignan substances had an effect on BP control or hypertension [67]. A study in the Netherlands reported that dietary lignan intake was associated with a lower prevalence of hypertension and lower BP [67]. Randomized clinical trials also reported that administration of lignan supplementation significantly reduced DBP, thus confirming that dietary lignans were effective in controlling blood pressure [68].

Direct evidence to support our finding for enterolactone could not be found in previous studies because studies for BP or hypertension using lignan metabolites are sparse, but a cross-sectional study showed a similar direction in association despite of non-significant result. In the U.S. National Health and Nutrition Examination (NHANES), it was reported that high BP had an inverse association with urinary enterolactone concentration, whereas it had a positive association with urinary enterodiol concentration [69]. This result was observed only for the result mutually adjusted for enterolactone, enterodiol and fiber (OR in the result mutually adjusted for enterolactone, enterodiol and fiber, OR = 0.67 (95% CI 0.44–1.01), OR = 1.58 (95% CI 1.07–2.34), respectively). In contrast, when they were not corrected for each other, the direction of the association was the same, but the result was more toward the null association.

Lignans are derived from a wide variety of plant foods, including vegetables, fruits and whole grain foods [70]. They are converted by intestinal-specific bacteria into the entertolignans, enterodiol, and enterolactone, and also the enterodiol is converted to enterolactone [71]. Although there is no direct evidence for vasodilation, it is reported that enterolactone has a higher antioxidant potency against lipid peroxidation than enterodiol [72]. In addition, the above-mentioned US state-run results suggest that enterolactone and enterodiol have a contradictory role in BP regulation, which raises the question of whether they will have the opposite effect, since they both have estrogenic properties. Given the lack of direct evidence for enterolactone in BP control and the rasing question, further studies are required.

There are several limitations in our study. First, we cannot establish the causality between plasma phytoestrogen and hypertension. Further studies, especially those with longitudinal study design, are warranted in order to confirm the results. A second limitation is that the plasma concentration of phytoestrogen was measured once at the baseline of subjects’ recruitment. We collected samples after overnight fasting so that we could minimize the effect of fluctuations in phytoestrogen concentration due to different time lapses after food intake. This limitation may be somewhat overcome because it was reported that isoflavone biomarkers could remain relatively stable, and phytoestrogen concentrations from spot sample could reflect the concentration of phytoestrogen in the body over one year period [73] although the half-life of phytoestrogen in the body is short (3~10 h) [74]. The third limitation is that we could not adjust for total calories and the other dietary factors due to a limited dietary information, although dietary soybean consumption may have correlation with healthy dietary habits. Fourthly, we were unable to conduct subgroup analyses according to menopausal status or other risk factors in the present study since the sample size was small. Despite these limitations, there are several strengths in this study. Firstly, the direct measurement of plasma phytoestrogen concentrations not only provides a metric of intake, but also reflects the results of the absorption and metabolism of isoflavones. Previous studies mainly used a food frequency questionnaire to measure soybean or isoflavone intake. Because the food frequency questionnaire (FFQ) is dependent on the subject’s memory, it is generally sensitive to measurement error. Therefore, plasma phytoestrogen levels may reflect relevant biological doses more accurately than the FFQ does [75]. Secondly, our study population was made up of Asian people who had individual variability of soybean intake. Total daily and energy-adjusted isoflavone intakes are known to differ by race and ethnicity. For example, the plasma levels of genistein were about 260 nmol/L in Japanese and Korean people, 34 nmol/L in Scottish people and 20 nmol/L in English people [76,77]. In Western populations, consumption of isoflavone may be too small to gain health effects from soybean, and the health effect of isoflavone may be underestimated or masked. Thus, we found a significant association between high plasma phytoestrogen concentration (equol and enterolactone) and low risk of hypertension, although our sample size was small. Thirdly, it is different from the aforementioned study because that study included the prevalent cases of hypertension, whereas our study included only incident hypertension and pre-hypertension cases after excluding subjects with taking antihypertensive medication at the time of cohort enrollment. Fourthly, this study included subjects with pre-hypertension stage, and also confirmed the preventive effect of equol and enterolactone in the prehypertension. This is an important clue that can prevent hypertension itself.

## 5. Conclusions

In conclusion, this study suggests that the specific isoflavone and the specific lignan, equol and enterolactone, can contribute to control prehypertension and hypertension. Further longitudinal studies are warranted to confirm the causal relationship be-tween plasma phytoestrogen and prehypertension. In addition, based on the natural history of hypertension and our finding that equol and enterolactone play a role from the prehypertensive stage, those substances might contribute to the prevention of hypertension and ultimate reduction in cardiovascular disease.

## Figures and Tables

**Figure 1 nutrients-13-04366-f001:**
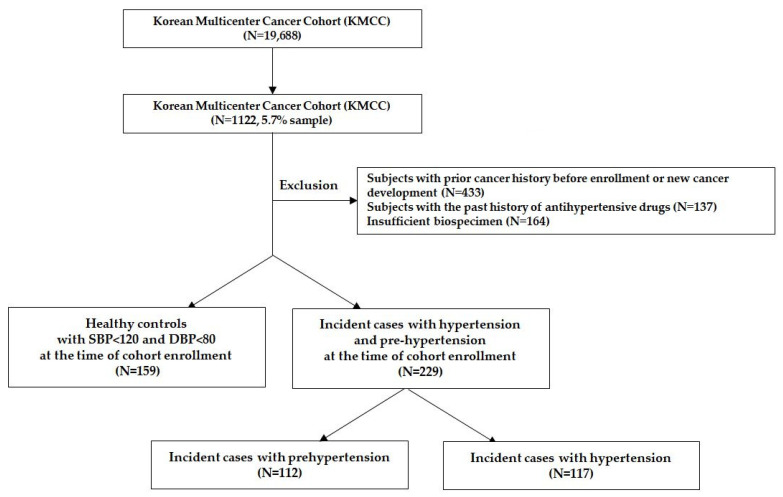
Study subjects to assess the association between plasma phytoestrogens and hypertension in the Korean Multicenter Cancer Cohort (KMCC).

**Table 1 nutrients-13-04366-t001:** Baseline characteristics of study population in the Korean Multicenter Cancer Cohort (KMCC) study, 1993–2004.

	Healthy Controls (*n* = 159)	Hypertension ^1^ including Prehypertension (*n* = 229)	*p*-Value
Age(years), mean (SD)	62.5 (7.5)	63.9 (8.9)	0.09
Sex, *n* (%)			0.40
Men	104 (65.4)	159 (69.4)	
Women	55 (34.6)	70 (30.6)	
Enrollment year, mean (SD)	1996.3 (1.8)	1996.0 (2.4)	0.15
Educated years, *n* (%)			0.63
≤6 (≤Elementary school)	45 (28.3)	67 (29.3)	
7–12 (Middle–High school)	111 (69.8)	155 (67.7)	
≥13 (≥University)	2 (1.3)	6 (2.6)	
Diabetes history, *n* (%)			0.54
Without diabetes	150 (95.0)	208 (94.1)	
Diabetes	7 (4.4)	13 (5.9)	
Smoking history, *n* (%)			0.26
Non-smokers	66 (41.5)	82 (35.8)	
Cigarette smokers	93 (58.5)	147 (64.2)	
Alcohol drinking history, *n* (%)			<0.01
Non-drinkers	79 (49.7)	80 (34.9)	
Alcohol drinkers	80 (50.3)	149 (65.1)	
Obesity, *n* (%)			<0.01
BMI < 25 kg/m^2^	138 (86.8)	173 (75.6)	
BMI ≥ 25 kg/m^2^	21 (13.2)	56 (24.4)	

^1^ Cases with high blood pressures were those with systolic blood pressure ≥ 120 mmHg or diastolic blood pressure ≥ 80 mmHg, no history of cancers, and who were not taking anti-hypertensive drugs at the time of cohort enrollment.

**Table 2 nutrients-13-04366-t002:** Association between serum phytoestrogen levels and incident prehypertension and hypertension in the Korean Multicenter Cancer Cohort (KMCC) study, 1993–2004.

Phytoestrogens (nmol/L)	Healthy Controls (*n* = 159)	Hypertension ^1^ including Prehypertension (*n* = 229)	
	*n* (%)	*n* (%)	OR (95% CI) ^2^
Isoflavones			
Genistein			
1T (<101)	53 (33.3)	64 (28.0)	1.00
2T (101–296.4)	36 (22.6)	86 (37.5)	1.49 (0.83–2.66)
3T (≥296.5)	70 (44.0)	79 (34.5)	0.80 (0.47–1.37)
*p*-trend			0.36
Daidzein			
1T (<57.5)	45 (28.3)	73 (31.9)	1.00
2T (57.5–208.9)	56 (35.2)	83 (36.2)	0.74 (0.43–1.30)
3T (≥209)	58 (36.5)	73 (31.9)	0.68 (0.39–1.19)
*p*-trend			0.11
Equol			
1T (<20.5)	36 (22.6)	81 (35.4)	1.00
2T (20.5–58.9)	53 (33.3)	76 (33.2)	0.54 (0.32–0.92)
3T (≥59)	70 (44.0)	72 (31.4)	0.34 (0.20–0.57)
*p*-trend			<0.01
Lignans			
Enterolactone			
1T (<31.6)	37 (23.3)	88 (38.4)	1.00
2T (31.6–76.35)	49 (30.8)	78 (34.1)	0.66 (0.37–1.17)
3T (≥76.4)	73 (45.9)	63 (27.5)	0.32 (0.18–0.57)
*p*-trend			<0.01

^1^ Cases were subjects who had systolic blood pressure ≥ 120 mmHg or diastolic blood pressure ≥ 80 mmHg, no history of cancers, and who were not taking anti-hypertensive drugs at the time of cohort enrollment. ^2^ adjusted for age, sex, enrollment year, education, smoking history, alcohol drinking history, and obesity.

**Table 3 nutrients-13-04366-t003:** Association between serum phytoestrogen levels and hypertension in the Korean Multicenter Cancer Cohort (KMCC) study, 1993–2004.

Phytoestrogens (nmol/L)	Healthy Controls (*n* = 159)	Prehypertension Cases ^1^ (*n* = 112)		Hypertension Cases ^2^ (*n* = 117)		
	*n* (%)	*n* (%)	OR (95% CI) ^3^	*n* (%)	OR (95% CI) ^3^	*p*-Ordinal
Isoflavones						
Genistein						
1T (<101)	53 (33.3)	31 (27.7)	1.00	33 (28.2)	1.00	0.67
2T (101–296.4)	36 (22.6)	51 (45.5)	1.91 (0.98–3.73)	35 (29.9)	1.08 (0.54–2.16)	
3T (≥296.5)	70 (44.0)	30 (26.8)	0.68 (0.35–1.33)	49 (41.9)	0.89 (0.48–1.65)	
*p*-trend			0.70		0.23	
Daidzein						
1T (<57.5)	45 (28.3)	36 (32.1)	1.00	37 (31.6)	1.00	0.69
2T (57.5–208.9)	56 (35.2)	42 (37.5)	0.75 (0.39–1.45)	49 (41.9)	0.87 (0.46–1.65)	
3T (≥209)	58 (36.5)	34 (30.4)	0.60 (0.31–1.18)	31 (26.5)	0.57 (0.29–1.13)	
*p*-trend			0.76		0.12	
Equol						
1T (<20.5)	36 (22.6)	40 (35.7)	1.00	41 (35.0)	1.00	<0.01
2T (20.5–58.9)	53 (33.3)	36 (32.1)	0.65 (0.33–1.27)	40 (34.2)	0.62 (0.32–1.19)	
3T (≥59)	70 (44.0)	36 (32.1)	0.50 (0.26–0.96)	36 (30.8)	0.42 (0.22–0.81)	
*p*-trend			0.01		0.04	
Lignans						
Enterolactone						
1T (<31.6)	37 (23.3)	44 (39.3)	1.00	44 (37.6)	1.00	<0.01
2T (31.6–76.35)	49 (30.8)	37 (33.0)	0.66 (0.34–1.30)	41 (35.0)	0.66 (0.35–1.29)	
3T (≥76.4)	73 (45.9)	31 (27.7)	0.38 (0.19–0.75)	32 (27.4)	0.28 (0.14–0.54)	
*p*-trend			<0.01		<0.01	

^1^ Pre-hypertension cases were those with systolic blood pressure of 120–139 mmHg or diastolic blood pressure of 80–89 mmHg, no history of cancers, and those who were not taking anti-hypertensive drugs at the time of cohort enrollment. ^2^ Hypertension cases were subjects who had systolic blood pressure ≥ 140 mmHg or diastolic blood pressure ≥ 90 mmHg, no history of cancers, and who were not taking anti-hypertensive drugs at the time of cohort enrollment. ^3^ Adjusted for age, sex, enrollment year, education, smoking history, alcohol drinking history, and obesity.

**Table 4 nutrients-13-04366-t004:** Association between serum isoflavone levels and hypertension according to obesity in the Korean Multicenter Cancer Cohort (KMCC) study, 1993–2004.

	Non-Obese (BMI < 25 kg/m^2^)			Obese (BMI ≥ 25 kg/m^2^)		
Phytoestrogens (nmol/L)	Healthy Controls (*n* = 138)	Hypertension ^1^ Including Prehypertension (*n* = 173)		Healthy Controls (*n* = 21)	Hypertension ^1^ Including Prehypertension (*n* = 56)	
	*n* (%)	*n* (%)	OR ^2^ (95% CI)	*n* (%)	*n* (%)	OR ^2^ (95% CI)
Isoflavones						
Genistein						
1T (<101)	49 (35.5)	48 (27.7)	1.00	4 (19.1)	16 (28.6)	1.00
2T (101–296.4)	27 (19.6)	64 (37.0)	1.69 (0.87–3.27)	9 (42.9)	22 (39.3)	0.49 (0.11–2.18)
3T (≥296.5)	62 (44.9)	61 (35.3)	0.86 (0.47–1.55)	8 (38.1)	18 (32.1)	0.51 (0.12–2.22)
*p*-trend			0.55			0.41
Daidzein						
1T (<57.5)	39 (28.3)	56 (32.4)	1.00	6 (28.6)	17 (30.4)	1.00
2T (57.5–208.9)	49 (35.5)	50 (28.9)	0.49 (0.25–0.94)	7 (33.3)	23 (41.1)	1.24 (0.31–4.95)
3T (≥209)	50 (36.2)	67 (38.7)	0.71 (0.38–1.34)	8 (38.1)	16 (28.6)	0.72 (0.19–2.70)
*p*-trend			0.33			0.57
Equol						
1T (<20.5)	34 (24.6)	60 (34.7)	1.00	2 (9.5)	21 (37.5)	1.00
2T (20.5–58.9)	44 (31.9)	50 (28.9)	0.63 (0.33–1.21)	9 (42.9)	26 (46.4)	0.30 (0.06–1.67)
3T (≥59)	60 (43.5)	63 (36.4)	0.63 (0.34–1.12) ^3^	10 (47.6)	9 (16.1)	0.06 (0.01–0.35) ^3^
*p*-trend			0.14			<0.01
Lignans						
Enterolactone						
1T (<31.6)	34 (24.6)	59 (34.1)	1.00	3 (14.3)	29 (51.8)	1.00
2T (31.6–76.35)	42 (30.4)	60 (34.7)	0.80 (0.42–1.52)	7 (33.3)	18 (32.1)	0.27 (0.06–1.29)
3T (≥76.4)	62 (44.9)	54 (31.2)	0.46 (0.24–0.88) ^3^	11 (52.4)	9 (16.1)	0.07 (0.13–0.34) ^3^
*p*-trend			0.01			<0.01

^1^ Cases were subjects who had systolic blood pressure ≥ 120 mmHg or diastolic blood pressure ≥ 80 mmHg, no history of cancers, and who were not taking anti-hypertensive drugs at the time of cohort enrollment. ^2^ Adjusted for age, sex, enrollment year, education, smoking history and alcohol drinking history. ^3^ Statistically significant *p*-value for heterogeneity between the ORs (95% CIs) in non-obesity and the ORs (95% Cis) in obesity group: *p* < 0.01 for equol; *p* < 0.01 for entereolactone.

## Data Availability

Data described in the manuscript, code book, and analytic code will be made available upon request pending.

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
