# Peer review of "The Association between Plasma Concentration of Phytoestrogens and Hypertension within the Korean Multicenter Cancer Cohort"

_nutrients, 2021, doi:10.3390/nu13124366_

Round 1

Reviewer 1 Report

In this study, logistic regression models to estimate a risk of hypertension were prepared, then association between plasma level of phytoestrogens and the risk was analyzed, and odds ratio of hypertension between tertile plasma phytoestrogens levels in hypertensive and healthy subjects was obtained. As a result, it is revealed that plasma levels of phytoestrogen metabolites, equol and enterolactone, were significantly associated with reduction of hypertension risk. The association was significance on obese compared to non-obese status, suggesting importance of the phytoestrogen metabolites on preventing hypertension in obesity.

It is estimated that two types of logistic regression models were created. The validity of models should be discussed by showing partial regression coefficients, significance, and fitness. A measuring method of plasma enterolactone should be added in “Measurement of plasma phytoestrogens”. In order to evaluate the antihypertensive effect of the phytoestrogens, interaction with the major blood pressure control systems in mammals, catecholamine and renin-angiotensin systems, should be discussed.

Author Response

We would like to thank you for the valuable comments from the Nutrients” and for taking time to review our article (nutrients-1445365).

We have made some corrections and clarifications in the manuscript after going over the reviewer’s comments.

Reviewer 2 Report

The paper deals with The association between plasma concentration of phytoestrogens and hypertension within the Korean Multicenter Cancer Cohort. Please see my suggestions below.

In the Abstract, L23. The authors are talking about liquid chromatography mass spectrometry. In subsection 2.3., L117-121 is about "time-resolved fluoroimmunoassay and gas chromatography-mass-spectrometry". Please clearly explain which method/methods was/were used fo rphytoestrogens determination. Furthher more, in L117-120 the terminology "time-resolved fluoroimmunoassay " was used 3 times, please reshape this part.

Introduction. L62. You must to add/complete some information about the use of these compounds in other disorders / diseases (i.e., menopause with all its stages, urinary incontinence, bone resorption, etc.), to better highlight the possibilities of their therapeutic involvement. In these regard please check Tit et al. Somatic-vegetative symptoms evolution in postmenopausal women treated with phytoestrogens and hormone replacement therapy. Iran. J. Public Health 2017, 46(11), 1128-1134; Bumbu A., et al. The effects of soy isoflavones and hormonal replacing therapy on the incidence and evolution of postmenopausal female urinary incontinence. Farmacia, 64(3), 2016, 419-422; Tit et al. Effects of the hormone replacement therapy and of soy isoflavones on bone resorption in postmenopause. J. Clin. Med., 7, 2018, 297. https://doi.org/10.3390/jcm7100297

L69-72. Please develop more the aim of the study, underlining better/more the novelty character of your research, the reasons for choosing this specific topic, the special aspects that make this paper relevant.

2. Materials and Methods. L77. If the subjects were recruited from 1993 to 2004, why publishing it now? Retoric questions.

Figure 1 is blurred. Please replace it with a better quality one (do not save it in any format but print screen the original figure)

Table 1. Please complete the head of the table for the first column (baseline characteristics? parameters?). Please check all tables to have a complete head of the have, for each column.

Conclusions.The authors should clearly explain how they deduced the chemopreventive effect for equol and enterolactone from the determinations performed.

Author Response

(The authors gave the same response as above.)

Reviewer 3 Report

"Our result for stronger association in the obese group was also similarily found in a meta-analysis of randomized controlled trials" It would be interesting if the authors considered how the  relationship between obesity and hypertension was also observed in tumors with an endocrine component ( and direct estrogenic control) such as prostate cancer, also reiterating how these components are essential in the development of metabolic syndrome also. It is suggested to refer to the following bibliography in support :

- N.S Murthy , S Mukherjee, G Ray, A Ray  Dietary factors and cancer chemoprevention: an overview of obesity-related malignancies J Postgrad Med 2009;55(1):45-54. 

- R. Asmar , J L Beebe-Dimmer, K Korgavkar, G R Keele, K A Cooney Hypertension, obesity and prostate cancer biochemical recurrence after radical prostatectomy. Prostate Cancer Prostatic Dis . 2013 ;16(1):62-6. 

- S. Di Francesco , I. Robuffo , M. Caruso , G. Giambuzzi , D. Ferri , A. Militello  9, E. Toniato Metabolic Alterations, Aggressive Hormone-Naïve Prostate Cancer and Cardiovascular Disease: A Complex Relationship. Medicina (Kaunas) 2019 ;55(3):62. 

-S. Di Francesco , R. L Tenaglia Metabolic Syndrome and Aggressive Prostate Cancer at Initial Diagnosis Horm Metab Res . 2017 Jul;49(7):507-509. 

Author Response

(The authors gave the same response as above.)

Round 2

Reviewer 1 Report

The indications have been properly revised.

Reviewer 2 Report

All the reccomandations have been respected.